# Rheological Properties of Industrial Hot Trub

**DOI:** 10.3390/ma14237162

**Published:** 2021-11-24

**Authors:** Marta Stachnik, Monika Sterczyńska, Emilia Smarzewska, Anna Ptaszek, Joanna Piepiórka-Stepuk, Oleg Ageev, Marek Jakubowski

**Affiliations:** 1Faculty of Mechanical Engineering, Koszalin University of Technology, 75-453 Koszalin, Poland; marta.stachnik@tu.koszalin.pl (M.S.); monika.sterczynska@tu.koszalin.pl (M.S.); emilus091@tlen.pl (E.S.); joanna.piepiorka@tu.koszalin.pl (J.P.-S.); 2Faculty of Food Technology, University of Agriculture in Krakow, 30-239 Kraków, Poland; anna.ptaszek@urk.edu.pl; 3Department of Food and Refrigeration Machines, Kaliningrad State Technical University, 236022 Kaliningrad, Russia; oleg.ageev@klgtu.ru

**Keywords:** hot trub, thixotropy, rheology, non-newtonian fluids

## Abstract

The boiling of beer wort with hops results in the formation of a hot trub, a sediment consisting mainly of water-insoluble tannin and protein conglomerates and hop residue. Hot trub is a waste product, removed in a clarifying tank and discarded. The use of barley malt substitutes in recipes for beer is associated with an increase in the amount of generated hot trub. In presented study, an analysis of the rheological properties of industrial hot trub was carried out. Samples varied with regard to the quantities of unmalted barley (0%, 35%, and 45%) and worts’ extract (12.5, 14.1, 16.1, and 18.2 °Plato) in the recipe. The rheology of each type of sludge was determined using a hysteresis loop at four different temperatures. The results showed the shear-thinning and thixotropic properties of the hot trub. It was found that, regardless of the raw material and extract used, all samples exhibited the same rheological properties, but with different values. It was also proved that both raw material composition and temperature affected the hot trub’s rheology. The highest values of viscosity were identified for malted barley, whereas the lowest apparent viscosity values were recorded for the hot trub with a 30% addition of unmalted barley. The Herschel–Bulkley model had the best fit to the experimental data.

## 1. Introduction

The essential ingredients for making beer are water, malt, hops, and yeast. Hopping involves boiling the wort with hops for a certain period. An important phenomenon that occurs during this time is the precipitation of proteins and polyphenols in the form of the so-called hot trub. From 5 to 30 min after boiling, bright floating flocs appear in the wort [1,2]. The particle size of the flocs varies from 30 to 80 µm [3], and up to 200 µm [4]. Recent research showed that these particles can even reach a diameter of 500 µm. The largest number of particles are of size of 30 to 140 µm [5], and the maximum diameter is estimated to be approximately 8000 µm [6]. If whole hop cones are utilized, the weight of the sludge after clarification will vary from 0.7 to 1.4 kg hl^−1^. If hop pellets are used, the weight of the hot trub alone ranges from 0.21 to 0.28 kg of wet weight per hl of wort and contains 80–85% water [7]. Industrial sludge after clarification usually contains approximately 75% wort and 25% dry matter. According to Narziß (1992), the amount of precipitated breakthrough ranges from 0.02 kg hl^−1^ to 0.08 kg hl^−1^ of wort [8]. The hot trub contains approximately 50–60% proteins, 20–30% tannins, 15–20% resins, 2–3% ash, and 1–2% fatty acids [9]. A good breakthrough is traditionally considered by brewers to be a good quality indicator for beer as it removes many unwanted substances from beer, including pesticides [10].

Since around 1960, a whirlpool has been used to remove residue. This is an empty standing cylindrical tank. The wort is pumped tangentially in the tank, forcing the liquid to rotate [11]. This creates the so-called cup of tea paradox; the residue settles in the form of a cone in the middle of the tank’s bottom [12]. Hot trub is removed with a significant amount of water. A cost-effective brewery uses between 4 and 7 hL of water per hL of ready beer [13]. This includes not only the water used to brew the beer itself but also the process water. In many studies, water usage is given for the whole brewery, and it ranges from 1.7 to 2.6 hL of water per hl of beer [14]. The World Bank group in 1998 published a report of water usage by the German brewing industry. The report revealed that from unfermented wort to whirlpool, 2 hL of water per hl of ready beer is used. Some sources reported that at a medium-sized brewery 800 hl of water per week is used for the removal of all kinds of deposits (including hot trub) [15]. Kopeć et al. (2020) analyzed compost made with hop trub and spent hops [16]. Mathias et al. (2015) suggest that the composition of hot trub is suitable as an additive in fermentation media [17]. This implies the need for new solutions for cleaning the whirlpool and for a valorization of bioresources through the recovery of valuable compounds from food waste [18,19].

Hot trub, despite its valuable nutritional properties (high protein content), is not suitable for feeding animals mainly due to its unpleasant bitter taste and possible high content of pesticides or mycotoxins [20,21,22]. Colloidal turbidity is mainly formed by proteins and polyphenols [23]. The turbidity-forming proteins contain a high proportion of the amino acid proline, to which polyphenolic compounds attach [24,25]. The bonds are formed due to intermolecular van der Waals interactions, reinforced by hydrogen bonds between the carbonyl group in proline monomers and the hydroxyl groups in phenolic compounds. Ionic and covalent bonds may also occur. The combination of proteins and polyphenols is initially soluble; however, when the complex reaches sufficient size, it becomes insoluble [26]. Hydrogen bonds are formed due to the charge differences of the compounds: polyphenols are positive, and proteins are negative. Flocculation decreases the protein’s electrical charge and increases its molecular weight [27,28]. The residue is considered a waste material in breweries. For 1 m^3^ of ready beer, 51.2 kg of solid waste is produced, including hot trub. The biochemical oxygen demand (BOD_5_) for wet residue is approximately ~85,000 ppm [29]. By comparison, a river is considered highly contaminated if its BOD_5_ exceeds 8 mg L^−1^, and the BOD_5_ of untreated wastewater is approximately 600 mg L^−1^. With current ecofriendly trends, it is crucial to find ways to repurpose waste, particularly such valuable waste. Okeyinka et al. (2019) suggested using brewery sludge residue ash as a base material for geopolymer binder [30]. The results of their study showed that it has good potential for this application.

Hop sediments can find use in cosmetology or medicine due to the presence of bioactive compounds [31] and their antioxidant and antimicrobial properties [32]. However, hot trub is still treated as waste and discarded [33]. Few options for brewing waste utilization are found in the literature. The most promising application is fertilizer in agriculture [34,35]. Another option is the co-biodrying of hot trub and the solid fraction of municipal waste [36]. Hot trub has a high content of sesquiterpenes; thus, it might be used to produce natural and cheap pest repellents for food storage [34]. Tesio et al. (2020) reported the production of lithium-sulfur battery cathodes by pyrolyzing the carbonaceous material contained in hot trub. When a high sulfur content (70%) was added to the carbon from this bio-waste using a “melt diffusion” method, a sulfur-carbon composite was formed and used as cathodes in Li-S batteries [37]. As mentioned before, the worst way to dispose of sediments from the brewery is to direct them to a municipal sewage system. This increases the costs of wastewater treatment and is unfavorable from an ecological and economic point of view [38].

Both the recycling of hut trub and its removal from wort require knowledge of rheological properties. The main reason to carry out rheological measurements is to find out the properties of materials under shear flow conditions, i.e., during such operations as pressing, mixing, or dosing. In the case of heterogeneous systems, an additional problem is the choice of a measurement methodology, as well as the selection of an appropriate model to describe the rheological properties. He et al. (2001) analyzed the effect of Trinidad Lake Asphalt (TLA) on the rheological properties of traditional petroleum bitumen [39]. Kim et al. (2019) assessed the rheological characteristics of hydrogen-fermented food waste [40]. Hydrogen-fermented food waste showed lower values for selected parameters than anaerobic digester sludge. It was established that waste with a lower viscosity value required less energy for agitation (by 30–67%) to ensure turbulent conditions (complete mixing). Malczewska and Biczyński (2017) studied municipal sludge [41]. The study was performed in a coaxial cylinder and rotating torque of the Couette–Searle type. Sludge concentration ranged from 4.40% to 2.09%. The experimental data for shear stress as a function of shear rate were fitted to the Herschel–Bulkley model. Cao et al. (2016) studied different rheological behaviors for sludge, with and without anaerobic digestion [42]. The results showed that the samples had shear-thinning and thixotropic properties. The Ostwald de Vaele model best fitted the experimental data. Liu et al. (2012) studied the rheological properties of coal–sludge slurry (a mixture of municipal wastewater sludge with coal, water, and additives). Coal–sludge slurry was a shear-thinning fluid with a thixotropic response, as described by the Herschel–Bulkley model [43]. In these studies, viscosity is used to optimize energy use for mixing and pumping, or as a control parameter of sludge processing.

This study intends to establish a rheological characterization of industrial hot trub obtained from brewing beer with different, unmalted barley substitutes. The impact of raw material and temperature on apparent viscosity is also examined. In addition, knowledge of rheological properties is necessary for the design of more efficient whirlpools in regards to different material properties and the simulation of the flow during hot trub removal.

## 2. Materials and Methods

### 2.1. Sample Preparation

Samples of hot trub (Figure 1a) were obtained from a medium-sized brewery with a typical European production profile of lager beers [44]. Wort was clarified in a whirlpool tank with a capacity of 530 hL made by ZIEMANN HOLVRIEKA, Ludwigsburg, Germany, in the year 2000. The vat was originally designed and made as a whirlpool-kettle, then converted into a rotatory tank. The brewhouse production is about 1,000,000 hL of beer per year. The barley and barley malt are two-row spring varieties with 10% protein content. Hops were granulated, T-90; bittering hops contained 14% α-acids, and aroma hops contained 3.5% α-acids. The samples were obtained from brews with full malt and with unmalted barley grain substitution (Table 1). Hot trub removed from all-malt wort was a control sample. Two samples from the same composition of raw materials had different extracts. It allowed us to verify if the sugar content had an effect on the rheological properties of the sludge. Clarified worts were also collected and their viscosity curves were recorded.

Approximately 1.4 kg of hot trub was collected from each brew. Initially, the sludge was mixed, and the water content for each type was determined using the MA 50R moisture analyzer (Radwag, Radom, Poland). Obtained material from the brewery differed in water content; thus, in all four hot trubs, it was adjusted to 76%. Otherwise, it would have been impossible to compare the viscosity values from the sample with 72% water content to the viscosity values of the sample with 80% water content. In literature, 76% water content predominates, even though some studies show a higher percentage [7,8]. In a measuring cup, a single sample of 46 g of hot trub was weighed, giving a total of 32 individual samples of each sludge. Cups were closed with a cap and left for 12 h in a refrigerator (to prevent mold) to relax.

### 2.2. Experimental Setup

Rheological measurements were performed with the HAAKE Viscotester iQ Air oscillatory rheometer (Thermo Scientific, Bremen, Germany).

#### 2.2.1. The Determination of Yield Stress and Flow Properties of Hot Trub

Hot trub has a relatively low dry matter content; however, its consistency is pasty (Figure 1b). Due to its unique nature, the first step of the rheological study was to determine the presence of yield stress. Linearly increasing stress was applied to the sample. Preliminary studies also included a selection of geometry and measurement parameters. Due to slipping, neither plate-plate nor plate-cone geometries could be used. For heterogeneous systems, such as foams [45,46], or suspensions, such as water solutions of starch [47], vane-cup geometry can be applied to measure yield stress and apparent viscosity. The selected system was validated by measuring the viscosity of glycerol (calibration fluid). The measuring procedure had to meet two criteria—maintenance of the laminar flow in the adopted geometry and such a value of the gap between the sensor and the bottom of the container with hot trub to guarantee the results are reproducible (5–10% deviation). The laminar flow was maintained for the shear rate value under 50 s^−1^. Exceeding this value caused transient and then turbulent flow and slipping of the sample. The diameter of the vane was 22 mm, and it had four plates; the cup had a diameter of 26 mm, which yielded a gap of 3 mm between the vane edge and the cup wall. There was no pressing out of the wort or squeezing of the deposit between the measuring geometries.

The viscosity curve was obtained on the base of stress determined from the hysteresis loop by continuously applying an increasing rate of strain from 0 to 50 s^−1^ for 100 s and decreasing from 50 to 0 s^−1^ for the same period of time. The time interval was chosen experimentally so that the hysteresis phenomenon could be captured at low shear rates. No pre-shearing was applied. It was noted that relaxed and pre-sheared samples exhibited up to 50% lower values of viscosity compared to samples that only underwent relaxation in measuring geometries (cups). Preliminary studies with shearing at a constant value for a specified time showed that the hot trub’s structure broke down within a few seconds and reached equilibrium value, depending only on the shear rate. This confirmed the presence of thixotropy. Additionally, since equilibrium was achieved within a few seconds there was no need for pre-shearing. Measurements were conducted at 20 °C, 40 °C, 60 °C, and 80 °C. As the sample cup was nonstandard, a water bath was used for heating. The rheological properties were measured after the center of the sample reached the appropriate temperature. The containers with hot trub samples during heating were closed to prevent water evaporation.

#### 2.2.2. The Determination of Rheological Properties of Wort

The wort’s viscosity was recalculated from the stress values measured in a double-gap concentric cylinder system. A flow curve was obtained for a shear rate range of 600 to 1000 s^−1^ at 20 °C. Temperature dependency was measured at a shear rate of
$\dot{\gamma }$ 1000 s^−1^ for a temperature range of 20 to 80 °C changing linearly by 0.2 °C/s.

### 2.3. Result Analysis

Each measurement was performed as three repetitions with a difference under 5%. The value of the yield stress of the hot trub was detected according to the procedure described in Yang et al. (2009 and 2011) [45,46]. This method consisted of loading the studied material with stress, which increased linearly over time, and observing the deformation (γ_0_). The value of the yield stress was defined as a point, where two fitting straight lines intersect in the (log (τ_0_)–log (γ_0_)) coordinate system. The values of the detected deformation were recalculated into shear rate, showed that, for higher than 0.5 s^−1^, hot trub started to flow.

We have attempted to describe the experimental data using one of the rheological models available in the literature. The best fit was determined through the highest value of R^2^. The Marquardt–Levenberg minimization procedure was used for the estimation of the parameters of the selected model [48]. Target function was formulated as:(1)$$\chi^{2}=\overset{N}{\underset{i=1}{\sum }}{\left(\eta_{i}-\eta \right)}^{2}\to min$$

The Herschel–Bulkley model (2) was successfully fitted to the experimental data obtained for a shear rate higher than 0.5 s^−1^. This model was chosen due to the characteristic shape of the flow curve and the presence of the yield stress. To ensure correct values of all three parameters, flow index n was estimated first and then yield stress and consistency index.
(2)$$\tau =\tau_{0}+k{\dot{\gamma }}^{n}$$

Additionally, statistical analysis was performed to determine the statistical significance of differences in the viscosity values of the analyzed hot trubs [43]. The mean value of the viscosity of each hot trub, from three repetitions, was statistically analyzed as a function of temperature and wort extract (recipe). Differences among means were estimated by analysis of variance ANOVA. Tukey’s HSD Test was performed to determine homogeneous groups.

## 3. Results and Discussion

Sterczyńska et al. (2021) [49] provided data from preliminary studies of sediment obtained on a semi-technical scale in laboratory conditions. Those studies focused on the influence of different malts and hopping times on the value of apparent viscosity. Qualitatively, the samples presented the same properties of shear-thinning and thixotropy as the hot trub from the industrial brewery. However, the values of those parameters were different.

### 3.1. Non-Newtonian Characteristics

The rheological properties of the sludge were evaluated based on viscosity curves in the form of hysteresis loops (Figure 2). Each sample showed a nonlinear relationship between viscosity and shear rate. Therefore, it can be concluded that the hot trub is a non-Newtonian fluid. As the shear rate increased, the hot trub tended to be less viscous, thus demonstrating shear-thinning behavior. Additionally, the presence of yield stress resulting from the roughness of hot trub particles was identified [50]. When the flow limit was exceeded, shear-thinning behavior appeared.

The viscous response at any given shear rate was reduced, showing that the hot trub became less resistant to flow. However, the viscosity increased at 80 °C. After each test, the water content of the samples was checked to exclude water evaporation and an increase in viscosity due to an increase in dry matter. The differences in viscosity values are most evident at high shear rates. Table 2 compares the maximum viscosity, viscosity of infinite shear rate, and recovery viscosity of the hot trubs and gives hysteresis loop area and yield stress values.

The highest viscosity values at each temperature were observed for hot trub 14, i.e., that which precipitated from all-malt wort. On the other hand, the lowest viscosity values were recorded for hot trub 12, i.e., that derived from the brew with 70% malted and 30% unmalted barley. The viscosity values of the hot trub from brews 16 and 18 were similar to those of hot trub 12. For each sludge, the viscosity limit at the shear rate of 50 s^−1^ decreased on average by 99% compared to the maximum values. The viscosity of the infinite shear rate of hot trub 14 at each temperature was approximately twice as high as that of the other hot trubs. The lowest viscosity values were observed at 60 °C. On the other hand, the highest decrease in the viscosity value was observed after heating the sludge from 20 °C to 40 °C (the highest decrease in η_max_ by 57% was observed for hot trub 14, while for the remaining sludges, a reduction in value by 40–44% was recorded). After heating to 60 °C, viscosity decreased by an additional 20%. At 80 °C, the viscosity of the hot trub increased to values close to η_max_ at 40 °C. The maximum apparent viscosities of hot trubs 12, 16, and 18 were 2% lower at 80 °C than at 40 °C, while the viscosity of hot trub 14 was 20% lower than that at 40 °C. No water loss was recorded in any sample, which is one of the possible explanations for the increase in apparent viscosity at 80 °C. The proteins forming conglomerates were denatured during wort boiling (and hot trub formation); thus, there was no denaturation during rheological measurement. Measurement at 80 °C was performed several times for the same sample, and the same result was always obtained, which ruled out the denaturation of proteins present in the suspension. A possible explanation for this is the swelling of the fibers of the heated hot trub, since the increase in the viscosity value was noticeable only at the initial shear. Moreover, the lowest values of the η_∞_ were identified at 80 °C.

In addition, the hot trub also showed yield stress. The highest values of τ_0_ were also observed for hot trub 14 and the lowest one for hot trub 12. Hot trubs 16 and 18 had very similar yield stress values. Similar to the maximum viscosity, the yield stress decreased with increasing temperature but increased at 80 °C. The highest reductions in τ_0_ values were recorded at 60 °C, by 60–75% of the value at 20 °C. At 80 °C, the yield stress value was 30–40% lower than that at 20 °C. Only for hot trub 14 were the τ_0_ values at 60 °C and 80 °C over 70% lower than those at 20° C.

The results of the statistical analysis are shown in Figure 3. Among the recipes at each analyzed temperature, hot trub 14 was statistically different from the other hot trubs. This sediment was collected from a beer brewed with 100% barley malt. There was no significant difference between hot trub 16 and 18. Those two sediments were brewed from the same composition of raw materials (55% barley malt + 45% unmalted barley) but had different extracts. However, the temperature had a significant influence on the apparent viscosity and yield stress value of each hot trub. It was noted that there was no significant difference in viscosity values at 40 and 80 °C. There was always a significant difference between viscosity values at 20 °C and at the higher temperatures.

Adjuncts lower both polyphenol and protein content in the wort, thus impairing hot trub formation. Another issue that arises due to the use of unmalted cereals is high amounts of β-glucans. Those compounds are broken down during malting. The presence of β-glucans increases the viscosity of the wort, thus slowing down mash separation and lowering extract recovery. Later, it slows down the filtration of beer and poses a risk of haze formation. These issues are associated with rye, buckwheat, oats, and wheat [51,52].

Studies on the physical properties of hot trub have been conducted previously. Jakubowski et al. (2015) used the Shadow sizing method to assess hot trub particle size for the same extract values as those in the above study [53]. All-malt wort and worts with 40% barley substitution were analyzed. All-malt wort had the highest number of particles with small diameters, which in this study resulted in a very high viscosity of the sludge. The investigated wort also had the greatest number of particles per volume. In the present study, the smallest number of particles was found in wort 12, which also explains the low viscosity of the hot trub. Hot trub is also characterized by a large variety of particle sizes, from very large to very small. Wort 16 and 18 had similar particle size and quantity distribution. The viscosities of these hot trubs were slightly higher than that of hot trub from wort 12.

Kunz et al. (2011) studied the influence of raw barley in the batch on the wort quality. Their results showed a noticeable increase in β-glucans for 25 and 50% barley proportion. The concentration of polyphenols and total nitrogen did not differ significantly between worts made with malt and brews made with unmalted grain [54]. It suggests that the difference in apparent viscosity values seems to depend on the particle size and the chain entanglement. A decrease in viscosity values as the emperature increased can be explained by the reduction in cohesive forces between molecules [55]. It can also be attributed to the shearing of a solvating layer from the long-chain molecules of the hot trub [56]. In a study by Senapati et al. (2010), the viscosity of fly ash suspension in water depended on solid fraction and increased with increasing particle size. Slurries also exhibited shear-thinning behavior. The authors also stated that with increasing particle size shear-thinning behavior is less evident [57]. A study by Konijn et al. (2014) confirmed those results. Moreover, they stated that particle size influences the viscosity of the suspension more if the liquid has a low viscosity value. Additionally, suspensions with same-sized particles had higher values of viscosity than did suspensions with varying particle diameters [58].

Yielding is present in flocculated suspensions of particles that exhibit mutual attraction. Interaction between floc creates a three-dimensional network present in the whole volume. Yield stress is thus expressed as a force per unit that is needed to breakdown the network. Hot trub has a fibrous structure, which can explain why it exhibits yielding [59,60].

### 3.2. Time-Dependent Decrease in Viscosity

Thixotropy was evaluated with the help of a qualitative test of the hysteresis loop, which refers to an area between upward and downward curves of shear rate ramps. Thixotropy refers to the decrease in viscosity when shear is applied and the ability of a material to rebuild its structure in time after the stress has been removed [61]. The loop area, and in consequence energy dissipated, changed with temperature and depended on the type of hot trub. The highest value of the energy dissipated was recorded for hot trub 14, and the lowest one for hot trub 12. Similar to the maximum viscosity value, energy dissipated ∆E decreased with temperature but increased when the sludge was heated to 80 °C. When the temperature increased from 20 °C to 40 °C, the largest drop in the energy dissipated (by 83%) was recorded in sludge 16, and the smallest drop (by 40%) in sludge 12. For sediment 14, a decrease by 45% was recorded, while in the case of sludge 18, a decrease by 72% was noted. After heating to 60 °C, the energy dissipated of hot trubs 12 and 14 decreased by another 40%, while that of a sludge 16 and 18 decreased by approximately 3%. When heated to 80 °C, the energy dissipated of sediments 16 and 18 increased to values similar to that at 40 °C, while for hot trubs 12 and 14, the loop areas were approximately 30% higher than ∆E at 40 °C.

In many suspensions, their structure strongly depends on the flow history. While at rest, a network forms, thus increasing viscosity; but when subjected to shearing, interparticle bonds are broken and viscosity decreases [62]. The development and breakdown of the suspension structure is a balance between particle collisions during the flow, flow stresses, Brownian force, and forces between particles. At rest, entanglement and attraction between particles are high, leading to high viscosity and elastic response. However, under the flow, particles are redistributed, detangled, and aligned, which lowers the viscosity of the suspension [63]. The suspension of non-spherical particles is usually characterized by large thixotropy. Such particles create a three-dimensional structure at much lower volume fractions than in the case of spherical particles [64].

The higher the value of the energy dissipated, the lower was the recovery of the structure. Hot trubs 14 and 18 had the highest ∆E values at all temperatures. The highest regeneration of the structure was observed for hot trub 12 at 20 °C. The low values of energy dissipated at higher temperatures resulted from greater destruction at the increasing shear rate, rather than from the reconstruction. Additionally, the values of maximum viscosity and viscosity on the return (η_rec_) were compared. Recovery viscosity decreased with the increase in temperature and did not increase at 80 °C. The largest difference in the maximum viscosity and recovery viscosity values occurred for hot trub 14 at 20 °C, for which η_rec_ was 88% η_max_. The lowest difference occurred for hot trub 12 at 60 °C, where η_rec_ was 38% η_max_ at this temperature.

### 3.3. Wort Viscosity

In the investigated suspension, the hot trub constitutes 24% and the remaining is wort. Wort is a Newtonian fluid (Figure 4a). The highest value of viscosity was found for wort 18 (2.5 mPa·s), while the lowest was for wort 14. It is worth noticing that wort with higher extract (14.1° P) had a lower value of viscosity (2.1 mPa·s) than wort with extract of 12.5° P and viscosity of 2.2 mPa·s. It is a small difference; however, it is consistent with other studies of the influence of unmalted grain on the viscosity of wort [64]. Wort 16 had a viscosity of 2.3 mPa·s. Figure 4b presents changes in viscosity with temperature. Temperature change from 20 to 80 °C caused a drop in viscosity value to, on average, 0.7 mPa·s.

### 3.4. Parameter Estimation for Herschel–Bulkley Model

For all investigated temperatures, it was found that the hot trub flow curves followed the same curvature. Several mathematical models are available to describe the relationship between viscosity and the shear rate of non-Newtonian fluids. The most adequate model was fitted using R^2^ and χ^2^ value (Equation (2)) as a minimization criterion. Models were fitted to the flow curves for a shear rate range from 0.5 to 50 s^−1^. First the value of yield stress was found, and then the other two parameters. Table 3 presents the parameters of the Herschel–Bulkley model that was successfully fitted to the experimental data (Figure 5).

The consistency index provides an idea of the viscosity of the fluid. However, to compare its values for different fluids, they should have a similar flow index (n) [65]. Analyzed hot trubs fulfil this requirement, given that value of n is almost constant (0.6 ± 0.1). It was noted that values of parameter k (consistency index) and n (the flow index) changed with temperature and varied among each hot trub. The highest values of these parameters were identified at 20 °C and the lowest at 40 °C. Simultaneously, the highest values of k and n were noted for hot trub 14, which was removed from a wort made with malted barley. These findings are consistent with the experimental data.

The rheological properties of hot trub will be used in computer simulations dealing with sedimentation in a clarifying tank. Early simulations dealt only with the flow of the wort in the presence of air, ignoring particles [55,66,67]. Later, the two-phase model was expanded with the third phase of the hot trub [68]. In that study, hot trub had the same viscosity as the liquid, which is an acceptable simplification. However, the rheological analysis clearly shows that the hot trub has a much higher value of viscosity and is a non-Newtonian fluid. Thus, these results will be part of an improved computer model. Moreover, knowledge of the rheological properties of the material is helpful for its preparation (transport, hydration, agglomeration, etc.) for possible recycling.

The 2020 Coronavirus pandemic had a widespread impact on all parts of society, including food production. Consumer interest in organic, so-called healthy, and functional foods increased rapidly during the lockdown. Following this demand, companies have delivered products with bioactive compounds. Waste products are thus seen as a source of such bioactives [69,70]. Erzinger et al. (2021) have discussed the antimicrobial properties of hot trub. The most promising compounds are antimicrobial β-acids and prenylated chalets, which have anti-cancer properties [71]. Omidiji et al. (2002) have discussed the successful enzymatic recovery of wort from cold trub. This is more impactful for the brewery, as they are interested in lowering the waste of wort [72]. It is quite obvious that hot trub is a valuable and versatile raw material, rather than a waste material.

## 4. Conclusions

In this article, the rheological properties of hot trub from an industrial brewing factory were examined. Four types of hot trub, made from three combinations of barley malt and unmalted barley were compared. Two of the analyzed hot trubs differed in the extract but were made from the same recipe (45% raw barley substitution). It was noted that the hot trub separated from all-malt wort significantly differed from the other hot trubs. It was also noted that neither the amount of added unmalted barley nor the extract value significantly influenced viscosity values. The hot trub exhibited yield stress which can be a result of its fibrous structure and interactions between polyphenol and protein. The hot trub’s behavior was similar to that of a shear-thinning system and showed a time-dependent nature. The highest value of energy dissipated was recorded for the hot trub taken from all-malt wort, whereas the lowest was recorded for the hot trub made with 30% barley substitution. The rheological properties were approximated using the Herschel–Bulkley model. The values of the *n* index described the non-Newtonian properties of hot trub, and *k* values indicated the consistency index. The presented results are most useful for computer simulations dealing with sedimentation in a clarifying tank. Additionally, the rheological properties of hot trub could be beneficial for designing bioactive compound extraction, determination of the mixing velocity in fermentation tanks, or for processing it into fertilizer.

## Figures and Tables

**Figure 1 materials-14-07162-f001:**
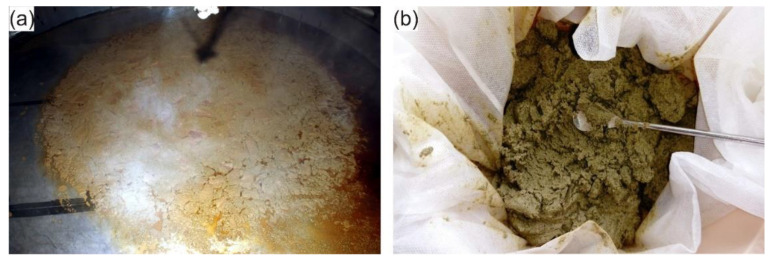
Hot trub (**a**) in the whirlpool tank; (**b**) batch of 70% malted + 30% unmalted barley, water content adjusted to 76%.

**Figure 2 materials-14-07162-f002:**
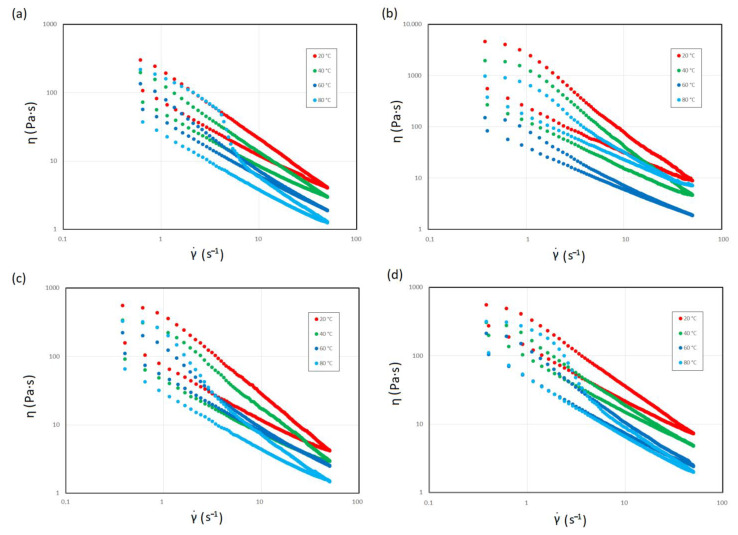
Viscosity curves of hot trubs as a function of shear rate at four temperatures: (**a**) hot trub from wort, 12.5° (70% malted + 30% unmalted barley); (**b**) hot trub from wort, 14.1° (100% malted barley); (**c**) hot trub from wort, 16.1° (55% malted + 45% unmalted barley); (**d**) hot trub from wort, 18.2° (55% malted + 45% unmalted barley).

**Figure 3 materials-14-07162-f003:**
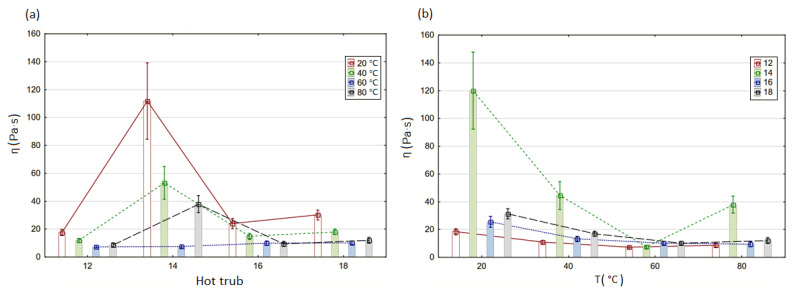
Graphs of mean viscosity values of hot trubs grouped by (**a**) recipe and (**b**) temperature (*n* = 3, α = 0.05).

**Figure 4 materials-14-07162-f004:**
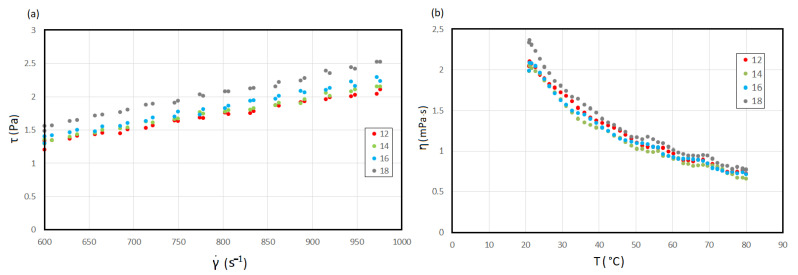
Beer worts: (**a**) viscosity curve; (**b**) viscosity as a function of temperature.

**Figure 5 materials-14-07162-f005:**
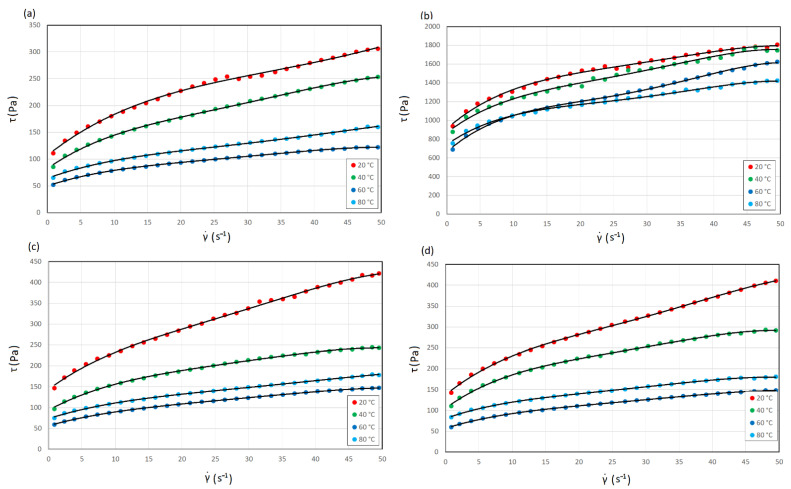
Herschel–Bulkley model fitted to flow curves of hot trubs as a function of shear rate at four temperatures: (**a**) hot trub from wort, 12.5°; (**b**) hot trub from wort, 14.1°; (**c**) hot trub from wort, 16.1°; (**d**) hot trub from wort, 18.2°.

**Table 1 materials-14-07162-t001:** Coding of hot trub samples.

Extract (°Plato)	Composition of Raw Materials	Code
12.5	70% malted + 30% unmalted barley	12
14.1	100% malted barley	14
16.1	55% malted + 45% unmalted barley	16
18.2	55% malted + 45% unmalted barley	18

**Table 2 materials-14-07162-t002:** Experimentally obtained values of limit, maximal, and recovery viscosity values, energy dissipated, and yield stress for hot trubs at different temperatures.

**η_max_ (Pa·s)**				
Hot trub	20 (°C)	40 (°C)	60 (°C)	80 (°C)
12	340.8	213.9	136.7	219.6
14	4630.0	1970.1	919.6	979.5
16	559.2	341.3	223.5	330.2
18	557.6	310.8	215.5	320.1
**η** ** _∞_ ** **(Pa·s)**				
12	4.1	2.9	1.9	1.3
14	9.1	4.8	9.8	7.3
16	4.2	3.0	2.4	1.5
18	7.4	4.0	2.5	2.0
**η_rec_ (Pa·s)**				
12	157.2	103.6	84.57	62.17
14	557.4	271.2	490.5	383.2
16	158.7	92.55	110.2	66.38
18	277.7	200.6	111.3	105.6
**ΔE (mJ)**				
12	20,410.4	12,284	2735.2	7469.6
14	140,960.8	76,675.2	25,230.4	28,922.4
16	39,473.6	6729.6	1802.4	6317.6
18	33,344.8	9292.8	8056.8	10,033.6
**τ_0_ (Pa)**				
12	116.6	78.7	55.1	67.4
14	975.6	960.1	758.2	878.1
16	137.4	88.7	62.5	73.7
18	148.6	107.4	66.2	81.2

**Table 3 materials-14-07162-t003:** Herschel–Bulkley model parameters for the tested hot trubs.

Hot Trub	T (°C)	*Herschel*–*Bulkley*				
		Parameters				
		τ₀ (Pa)	k (Pa·s^n^)	n (-)	χ^2^	R^2^
12	20	112.3	14.5	0.5	391.4	0.99
	40	76.7	34.0	0.5	259.3	0.99
	60	54.4	8.9	0.5	71.4	0.99
	80	66.6	13.4	0.6	243	0.99
14	20	982.3	322.9	0.7	645.4	0.99
	40	967.2	108.5	0.6	560.1	0.99
	60	767.5	72.8	0.4	717.2	0.99
	80	877.9	132.6	0.5	327.8	0.99
16	20	135.6	18.5	0.7	114.3	0.99
	40	89.5	19.2	0.6	528.2	0.99
	60	62.2	8.8	0.5	88.8	0.99
	80	73.3	18.2	0.6	118.7	0.99
18	20	146.8	40.7	0.7	126.1	0.99
	40	107.1	16.7	0.6	479.1	0.99
	60	66.06	12.0	0.5	67.3	0.99
	80	82.4	14.6	0.6	379.4	0.99

## Data Availability

Data sharing is not applicable to this article.

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
