# Peer review of "Rheological Properties of Industrial Hot Trub"

_materials, 2021, doi:10.3390/ma14237162_

Round 1

Reviewer 1 Report

Comments on the Manuscript MATERIALS-1165138 R2

Some conceptual mistakes still maintain in R2 version of the manuscript. They must be solved before giving a positive recommendation for publication in Materials.

1. Cross model (JCS 1965) was obtained imposing steady conditions for the measurement of viscosity. Authors should revise Cross publication. They will read refering to Cross equation (page 425), that “It should be emphasized that it refers to equilibrium values of viscosity”.

Thank you for the comment. We are aware that this model best represents equilibrium conditions, but faced with a situation where other models did not give interpretable parameters we decided to use this equation. Our main concern was to determine the K constants as measures of changes in material interactions in the context of temperature and its effect on sediment structure.

This argumentation should be included in the text to avoid misinterpretations. Authors should clarify that Cross model is not really used, but an equation with the same formal aspect. Consequently, it is questionable if the physical meaning of the parameters given by Cross is extensible to the equation used by the authors.

Power models do not reflect viscosity changes as a function of shear rate with a satisfactory value of the chi2 statistic. For this reason, we decided to use the Cross model as the best representation of viscosity changes induced by shear rate in the range from 0.5 to 50 s-1. We reasoned that, according to the interpretation of the exponent m, a deviation from Newtonian behavior would be demonstrated and the values of the parameter K would describe the effect of temperature on changes in rheological properties. Appropriate justification has been included, lines 213-232 and 363-378.

I insist in the fact that authors should express unambiguously that they HAVE NOT used Cross model. They have used an equation with the same mathematical aspect than that resulting from the treatment by Cross of STEADY viscosity curves (correct nomenclature because  is not apparent viscosity because this term is restricted to steady state values). I mean that, in authors case,  is not zero-shear viscosity (correct nomenclature) because this is a parameter defined in STEADY conditions in Cross´ model and in Rheology literature. Their , despite the same symbol is used, is a value corresponding to some unspecified transitory (virtually at rest) state. Similarly, while  in Cross´ model is the STEADY viscosity at infinite (high) shear rate, authors´ , although with the same symbol, is the viscosity in some unspecified transitory state at infinite (high) shear (correct nomenclature). In the abstract (lines 24-25) and in Conclusions (lines 451-452) authors insist they have used Cross´ model. This is not true because their results are not recorded at steady state, which is a condition imposed by Cross to obtain his equation. Simply, authors should say they have used an equation with the same mathematical format. Certainly, authors can affirm that their K is some time parameter (although correspond to a series of viscosity values obtained in a sequence of undefined microstructural states which, in my opinion, debilities its utility, because, in general, for different shear acceleration the results would be different), and m is a parameter defining the kind (increasing or decreasing viscosity with shear) and the rate (large or small value) of variation of viscosity with shear, but with the same ambiguity as indicated for K. Finally, Cross equation is not applicable to viscoplastic materials (infinite zero-shear viscosity) but authors affirm their materials show yield stress.

Note that I would not have done these comments if authors have used some different mathematical equations selected from those existing in literature but without any physical foundation. But they have just selected one based on physical grounds (Cross) that assumes STEADY conditions.

2. Authors identify shear-thickening behavior at very low shear rate values. They should note that shear-thickening and also shear-thinning are time-independent viscous behaviors, so, the apparent viscosity should be recorded at the steady state. It is illustrative that authors (acceptably) use “time-dependent shear-thinning” when they are talking about “thixotropy”. The “apparent shear-thickening” authors describe is probably the result of viscosity measurements in the transitory state. Or, perhaps, some artifact. By the way, what pre-shear was applied to samples before testing?

Thank you for the comment and for noting this element. No pre-shear was applied. Preliminary tests showed that pre-shearing significantly changed viscosity of the hot trub. Shear-thickening and shear-thinning properties are addressed in lines 292-301 and 320-325.

The fact that pre-shear changes author´s results is an evidence of that it is necessary to design a rheological test with pre-shear. The reason of this previous mechanical action on to the sample is to ensure that studies are controlled and reproducible. No answer has been given to my query on the “apparent” shear-thickening shown at low shear rates (Figures 2). In fact, authors maintain the message on the observation of shear-thickening (line 226).    

The conditions for rheological measurements were determined from numerous preliminary measurements. The problem of rheological measurements for systems of paste consistency is the lack of an unified test methodology, especially in the case of sediments. Standard plate to plate or cone to plate measurement systems did not work well in this case. Based on the experience of using the vane system for testing pasty starch suspensions and starch gels, it was decided to try this measurement system. The literature data indicate that the vane system is also useful for testing the yield stress of such "delicate" systems as food foams (Yang 2009). The results of the preliminary experiments showed the repeatability of the obtained results. The slurries we studied exhibit characteristic rheological properties; after exceeding the deformation corresponding to the limiting shear rate (about 1s-1), they show a shear-thinning phenomenon. The apparent viscosity decreases rapidly with increasing shear rate. We performed experimental tests based on the equilibrium experiment of Kemblowski and Petera, which involves measuring the change in shear stress over a given time interval at a given shear rate. The results of these experiments showed that the sediment structure (in the sense of cohesion), is destroyed within a few seconds, which is manifested a time constant value of viscosity, depending only on the shear rate. The results of this study confirm the thixotropy phenomenon manifested by the hysteresis of the flow curves. Moreover, it can be concluded that stationary conditions are reached in a time interval of a few seconds. For this reason, we decided not to use the pre-shear procedure. The time scale of the experiment (100 seconds for the shear rate increase from 0 to 50 s-1) was chosen in order to visualize the hysteresis phenomenon. Information about time scale is included in line 190- 196 and 249-252.

I agree with the argumentation that justify the use of vane in cup geometry. This was not my question. I am interested in reasons given by authors to avoid pre-shear (constant shear applied to sample before testing to eliminate hazardous results that can result from sample placing in the geometry (the cup in this case)). It is not clear why they decided not to apply pre-shear.

On the other hand, the existence of hysteresis loop indicates time-dependency (not necessarily thixotropy because this phenomenon demands reversibility, which, I think, authors have not confirmed). The problem with hysteresis loops is that the experimenter is seeing the simultaneous effect of shear rate and time. To simplify is indicated to record steady values because they only will depend on the corresponding shear rates. Shear-thinning and shear-thickening are TIME-INDEPENDENT viscous behaviors, therefore, is not appropriate to use these terms with hysteresis loops. Therefore, the erroneously named “shear-thickening” at low shear rates is probably due to the instant when the viscosity was recorded at those small shear rates. Note that in the reference [52] cited in the manuscript by authors (and others on disposal in literature) shear thickening is preceded by Newtonian or slight shear-thinning behavior and not related to yield stress (I recommend Barnes (1989) seminal paper). 

Reviewer 2 Report

The manuscript is interesting and supported by a lot of references, but there has to be done a many improvements.

I have the following suggestions:

  1. The authors should read a whole manuscript again and do detailed language proofreading and corrections. For example there are two identical sentences in Lines 148-150.
  2. Table 2 and Table 4; there are not standard deviations - means that all measurements were performed only once? It has to be measured at least in triplicates and the standard deviations, same as some statistical analysis have to be added in the manuscript.
  3. Same for the another results, there should be some statistical analysis, because then you can not say, that the founded differences between samples are statistical significant different, which is important for the scientific article.

Round 2

Reviewer 1 Report

Comments on the Manuscript MATERIALS-1165138 R2 or MATERIALS-1444378

I recommend publication in Materials after considering my last comment.

  1. Cross model (JCS 1965) was obtained imposing steady conditions for the measurement of viscosity. Authors should revise Cross publication. They will read refering to Cross equation (page 425), that “It should be emphasized that it refers to equilibrium values of viscosity”.

Thank you for the comment. We are aware that this model best represents equilibrium conditions, but faced with a situation where other models did not give interpretable parameters we decided to use this equation. Our main concern was to determine the K constants as measures of changes in material interactions in the context of temperature and its effect on sediment structure.

This argumentation should be included in the text to avoid misinterpretations. Authors should clarify that Cross model is not really used, but an equation with the same formal aspect. Consequently, it is questionable if the physical meaning of the parameters given by Cross is extensible to the equation used by the authors.

Power models do not reflect viscosity changes as a function of shear rate with a satisfactory value of the chi2 statistic. For this reason, we decided to use the Cross model as the best representation of viscosity changes induced by shear rate in the range from 0.5 to 50 s-1. We reasoned that, according to the interpretation of the exponent m, a deviation from Newtonian behavior would be demonstrated and the values of the parameter K would describe the effect of temperature on changes in rheological properties. Appropriate justification has been included, lines 213-232 and 363-378.

I insist in the fact that authors should express unambiguously that they HAVE NOT used Cross model. They have used an equation with the same mathematical aspect than that resulting from the treatment by Cross of STEADY viscosity curves (correct nomenclature because  is not apparent viscosity because this term is restricted to steady state values). I mean that, in authors case,  is not zero-shear viscosity (correct nomenclature) because this is a parameter defined in STEADY conditions in Cross´ model and in Rheology literature. Their , despite the same symbol is used, is a value corresponding to some unspecified transitory (virtually at rest) state. Similarly, while  in Cross´ model is the STEADY viscosity at infinite (high) shear rate, authors´ , although with the same symbol, is the viscosity in some unspecified transitory state at infinite (high) shear (correct nomenclature). In the abstract (lines 24-25) and in Conclusions (lines 451-452) authors insist they have used Cross´ model. This is not true because their results are not recorded at steady state, which is a condition imposed by Cross to obtain his equation. Simply, authors should say they have used an equation with the same mathematical format. Certainly, authors can affirm that their K is some time parameter (although correspond to a series of viscosity values obtained in a sequence of undefined microstructural states which, in my opinion, debilities its utility, because, in general, for different shear acceleration the results would be different), and m is a parameter defining the kind (increasing or decreasing viscosity with shear) and the rate (large or small value) of variation of viscosity with shear, but with the same ambiguity as indicated for K. Finally, Cross equation is not applicable to viscoplastic materials (infinite zero-shear viscosity) but authors affirm their materials show yield stress.

Note that I would not have done these comments if authors have used some different mathematical equations selected from those existing in literature but without any physical foundation. But they have just selected one based on physical grounds (Cross) that assumes STEADY conditions.

Thank you for the comment and for noting this element. We are grateful for the extensive explanations. We are aware that the Cross equation does not apply to fluids with yield stress. However, the conventional fitting of the Herschel–Bulkley model provided negative values for model parameters. On this basis, H-B model was rejected. The equation mimicking the Cross model had a satisfying fit, and for that reason, it was selected.  We have reviewed our data and literature and provided changes to the manuscript. We refitted available models to the flow curves. Instead of first estimating yield stress value and then the other two parameters, we first estimated value of n. This method ensured positive and correct estimation of yield stress and positive values of n and k parameters. Changes are included in the Summary, line 24; Nomenclature; Result analysis, lines 213-221; Parameter estimation, lines 387-399; Conclusions, lines 429-434.

You say that H-B model was rejected due to it supplies negative yield stress values but you say in the abstract that H-B model had the best fit to the experimental data (?). By the way, units of K-parameter in table 3 are Pa·sn not s. The problem with HB model is that K-values measured with different unities cannot be compared (despite extensive use in literature). In this line, I like your comment on lines 393-394 demanding the same n-value to compare K-values. Authors should enhance that their n-values were almost constant (0.6+0.1), which could justify why they could compare K of different suspensions. Other models like Casson or Modified Bingham can avoid this problem.

Author Response

Yellow highlighted sentences are the sentences and elements updated within the manuscript after the current round of reviews

Reviewer #1
Thanks for re-reviewing the manuscript, your comment was all valuable. Below you can find our replies to your comments and requests for changes. All changes in the main text are indicated in red. The answer is as follow:

When we were first fitting rheological models H-B, Casson and Bingham produced negative values, because we were fitting models to the viscosity curves. Moreover, we used method of finding all parameters simultaneously. Currently, we first found value of yield stress and then found other two parameters. That’s why now H-B has both R2 of 0.99 and positive parameters.

In line 394 we added suggested information, that n-values are similar and in table 3 we corrected units of k.

Reviewer 2 Report

All sugestions were accepted and corrected by authors. The manuscript can be accepted for publication.

Author Response

Thank you for re-reviewing the manuscript, your previous comments were all valuable.

This manuscript is a resubmission of an earlier submission. The following is a list of the peer review reports and author responses from that submission.

Round 1

Reviewer 1 Report

Manuscript is well written and the topic has high industrial and economic interest. 

Introduction is detailed and shows significance of the subject.

Experimental part of the manuscript focuses on rheological properties of hot trub.

Methods should be described more clearly: the viscosity and hysteresis parameters discussed in Results part are not defined in the methods session.

Figure 2. needs some clarification: in the text (Line 178) log-log system is mentioned, though every axis on the Figure seems to be a linear scale. For what reason occur in each figure (Fig.2. a,b,c,d) 8 curves, though there were 4 temperature level tested?

Figure 3.: Samples seem to be the same beacuse of the same indication signs. It is not clear at which shear rate the viscosity was measured.

Explanation of the measured results are clear and adequate.

In Conclusion the connection between the measured results and the residue recycling should be mentioned (since it was mentioned as aim of the study: Lines 132-134)

Reviewer 2 Report

I am glad to have opportunity for reviewing this manuscript. This manuscript verified rheological characteristics of hot trubs the residue came from brewing process. For understanding its characteristics and processing it to use in industry, the present study would be prerequisite. I think its research design and its result are well-designed; however, this manuscript has some critical vulnerable points in present form. I hope that my opinions could help improving the quality of your study, and my comments for your manuscript are as follows:

Title

  1. Your title contains major subjects and your purpose, but it is too concise. I think that if you add more important outputs of your study in title, it could improve attentions and help understanding your study.

Abstract

  1. line 15-16: The intended meaning of “(obtained from … wort’s extract)” is not clear, therefore you need to revise this sentence. I think that you want to present your sample conditions here, but it is too hard to comprehend. Therefore, please describe your sample groups in different ways for enhancing understanding.

  1. I think the definition of “trub” is not explained enough in abstract section. Its definition is well-revealed in line 40-41. Likewise, I recommend inserting the definition of trub in abstract section to enhance understanding of readers.

  1. line 19-20: I cannot understand what you want to claim here. “qualitatively” and “quantitatively” means which things, respectively?

After I peruse your whole manuscript, I conjecture that “qualitatively” means ‘non-Newtonian’, ‘thixotropic’, and ‘shear-thinning’ traits, and “quantitatively” could mean parameters of equations fitting to the Cross model in Table 3. However, I am not sure still. You must explain accurately.

  1. line 23: I think you missed a comma behind “unmalted barley”.

  1. line 23: Please insert “The” in front of “Cross model” because the “Cross model” is a proper noun.

Introduction section

  1. line 53: “a considerable amount of fats” means that there are lots of fats in hot trub but the content of fats is not presented here. If considerable amount of fats in hot trub, then please present its approximate content like other constituents in the sentence. Or you need to revise the statement, “a considerable amount”.
  2. line 65: You need to insert “soluble” in first phrase of the sentence, “The combination of proteins…”, to match the meaning of first phrase and second phrase.

  1. The 5th paragraph (line 101-110) should be removed for unifying flows of introduction.

  1. The 2nd paragraph (line 60-69) need to be moved to behind of 6th paragraph (line 111-129) and you need to refine flows of two paragraphs. Or you could remove the 2nd paragraph. I think that 2nd paragraph contains helpful information but it vitiates unity and flows of Introduction in this form.

M&M section

  1. Could you show the specification (size, product name, etc.) and manufacturer of whirlpool vessel?

  1. For enhancing reproductibility of your study, please write the source of your materials (malted barley, unmalted barley)

Results and discussion

  1. It is very fundamental question but I am so confused. I wonder about the state of your hot trub sample (with 76% water content). Can you call this sample as “hot trub”? Because the “hot trub” is sludge material without mixing with external water. Therefore, the condition of your samples are not same with “hot trub” and in this case, the name of your sample could be changed. I found the paper Sterczynska M et al., (2021) in Molecules, and the same problem was found in the paper (However, the case of this paper need to be set aside because one of author (Sterczynska M) participated in both studies). In other previous studies, they did not treat trubs as fluids because they are sludge state and they regarded trub as solid material. Therefore, they measured particle size and its size distribution. In my opinion, you need to rechristen your sample solution (containing hot trubs in it) through your manuscript. In your manuscript, sometimes you called your sample as ‘slurry’ and I think that ‘hot trub slurry’ could be suitable sample name but I am not sure.

  1. line 172: Please change “sediment” to other words. In this form, the “sediment” could mean sediment of your hot trub solution.

  1. line 180: “Plateau” should be changed to “plateau”. I searched related articles whether “Newtonian Plateau” is a proper noun; however, other studies using a “Newtonian plateau”, instead. In line 182-183 same errors are found.

  1. The subtitle is missing in line 171.

  1. In Table 2, significant figures (significant digits) are not matched and jumbled.

Discussion

  1. You have “3.Results and Discussion” section, but you also inserted “4. Discussion”. Please merge sections or please revise the head of 3rd section as “3.Results”. I recommend merging two sections.

  1. In your results and discussion sections, your results and related figures and tables are sophisticated and well-designed. It shows the rheological properties of your samples properly. However, the comparison of results with other previous studies or interpretation is so lacking. In addition, the possible reasons that your samples show thixotropic and shear thinning flows are missing, I think. Could you please add discussions? For example, concrete slurry shows same fluid characteristics (thixotropic and shear thinning) and there are other solutions also have same traits. They could be helpful to interpret your result.

Conclusions

  1. I think that it could increase significance of your study if you add the possible uses of your results in industry or science field in the future.

Reviewer 3 Report

Looking at your cited literature it is obvious to identify your expertise: 34 citation from papers of brewing technology, only 8 dealing with rheology. What is your motivation to investigate the rheological properties of hot trube?

line 115: What does it mean: lower values of rheological properties?

line 149: You explain that you adjusted the water content to 76%. Unfortunately you are modifying by this the rheological properties of the trub!

line 159: You have used a vane instead of a cylinder. How did you calculate the viscosity. Did you calibrate the system and if yes how did you do?

line 163: You performed shear rate ramps. Are you sure that the gradient was low enough you measure stationary stress values.

Fig 2: The shear thickening behaviour at low shear rates could be caused by thixotropic effects. Did you consider this?

line 199: What is the time scale of thixotropy? Why don't you show the hysteresis loop?

line 222: How did you identify yield stress? How large are the yield stresses?

Fig 3: For what shear does fig 3 hold?

line 243: The cross model gives a viscosity plateau for low and hi8gh shear rates. But clearly, your experiments do not show plateaus for low shear rates. So the question arise: Why did you choose the Cross model?

line 282: Your discussion is not a discussion but just a presentation of values without any explanation.

Reviewer 4 Report

Comments on the Manuscript MATERIALS-1165138

Authors present hysteresis loops of different hot trub samples varying temperature and composition.

Authors should correct some conceptual mistakes before this manuscript could be considered for publication in Materials.

  1. The existence of zero-shear viscosity is opposite to the existence of a yield stress value. However, authors calculate yield stress (I do not know how) but after that they fit data to Cross model, that considers zero-shear viscosity as one parameter.
  2. Cross model (JCS 1965) was obtained imposing steady conditions for the measurement of viscosity. Authors should revise Cross publication. They will read refering to Cross equation (page 425), that “It should be emphasized that it refers to equilibrium values of viscosity”.
  3. Authors identify shear-thickening behavior at very low shear rate values. They should note that shear-thickening and also shear-thinning are time-independent viscous behaviors, so, the apparent viscosity should be recorded at the steady state. It is illustrative that authors (acceptably) use “time-dependent shear-thinning” when they are talking about “thixotropy”. The “apparent shear-thickening” authors describe is probably the result of viscosity measurements in the transitory state. Or, perhaps, some artifact. By the way, what pre-shear was applied to samples before testing?
  4. Authors should avoid the use of “pseudoplastic”. It is an old expression for the currently accepted “shear-thinning”.
  5. In the definition of thixotropy (lines 200-201) authors should include the decrease of viscosity when a shear is applied followed by, certainly, the rebuild of the original structure when the shear is removed.
  6. Figure 3 refers to “wort viscosity” at different temperatures. But, at what shear rate? Is it Newtonian?
  7. Discussion and Conclusions parts are really a list or a summary of results.
  8. Nomenclature: unit of shear stress and yield stress is Pa. Why authors talk about “reduced” hysteresis area? Why is it “reduced”?

Round 2

Reviewer 2 Report

On the whole, I think that authors properly addressed suggestions of mine and other reviewers'. Thanks for your consideration and efforts about it.

However, I found some minor points should be revised.

1. In the abstract section, I could understand the meaning of "qualitatively" and "quantitatively"; however, it is not solved in manuscript. Please rewrite the sentence to make readers understand it. I think no one could understand their meaning in this form.

2. As the "3.Result and Discussion" and the "4.Discussion" section was merged, you must revise "5.Conclusion" with "4.Conclusion".

3. In my opinion, the manuscript needs appropriate English editing in this form.

Reviewer 3 Report

I'm not satisfied how you went around with my comments. The idea is that you consider the comments to modify the text. For the rheological part of the publication - which is the most important part of your publication, according to the title - you made only minor modification which do not satisfy me. 

Please go through my previous comments and modify your publication. 

Reviewer 4 Report

Comments on the Manuscript MATERIALS-1165138 R1

Some conceptual mistakes before this manuscript could be considered for publication in Materials.

  1. Cross model (JCS 1965) was obtained imposing steady conditions for the measurement of viscosity. Authors should revise Cross publication. They will read refering to Cross equation (page 425), that “It should be emphasized that it refers to equilibrium values of viscosity”.

Thank you for the comment. We are aware that this model best represents equilibrium conditions, but faced with a situation where other models did not give interpretable parameters we decided to use this equation. Our main concern was to determine the K constants as measures of changes in material interactions in the context of temperature and its effect on sediment structure.

This argumentation should be included in the text to avoid misinterpretations. Authors should clarify that Cross model is not really used, but an equation with the same formal aspect. Consequently, it is questionable if the physical meaning of the parameters given by Cross is extensible to the equation used by the authors.

  1. Authors identify shear-thickening behavior at very low shear rate values. They should note that shear-thickening and also shear-thinning are time-independent viscous behaviors, so, the apparent viscosity should be recorded at the steady state. It is illustrative that authors (acceptably) use “time-dependent shear-thinning” when they are talking about “thixotropy”. The “apparent shear-thickening” authors describe is probably the result of viscosity measurements in the transitory state. Or, perhaps, some artifact. By the way, what pre-shear was applied to samples before testing?

Thank you for the comment and for noting this element. No pre-shear was applied. Preliminary tests showed that pre-shearing significantly changed viscosity of the hot trub. Shear-thickening and shear-thinning properties are addressed in lines 292-301 and 320-325.

The fact that pre-shear changes author´s results is an evidence of that it is necessary to design a rheological test with pre-shear. The reason of this previous mechanical action on to the sample is to ensure that studies are controlled and reproducible. No answer has been given to my query on the “apparent” shear-thickening shown at low shear rates (Figures 2). In fact, authors maintain the message on the observation of shear-thickening (line 226).    

  1. Discussion and Conclusions parts are really a list or a summary of results.

Thank you for the comment and for noting this element. Discussion was provided across the text, lines 240-272; 292-300; 320-333; 399-420.

Conclusions part is really a list of results.
